# Rapid and inducible mislocalization of endogenous TDP43 in a novel human model of amyotrophic lateral sclerosis

Johanna Ganssauge[1,2], Sophie Hawkins[1,3†], Seema Chandramohan Namboori[1,3†], Szi Kay Leung[3], Jonathan Mill[3], Akshay Bhinge[1,3*]

[1]Living Systems Institute, University of Exeter, Exeter, United Kingdom; [2]Biosciences, University of Exeter, Exeter, United Kingdom; [3]Clinical and Biomedical Sciences, University of Exeter, Exeter, United Kingdom

## eLife Assessment

TDP-43 mislocalization is a key feature of some neurodegenerative diseases, but cellular models are lacking. The authors endogenously-tagged TDP-43 with a C-terminal GFP tag in human iPSCs, followed by expression of an intrabody-NES that targeted GFP to the cytosol. They **convincingly** report physical mislocalization and functional depletion of TDP-43, as measured by microscopy and RNAseq. This method will be **valuable** to investigators studying the biological consequences of TDP-43 mislocalization and the methodology is in line with the current state-of-the-art.

*For correspondence:
a.bhinge@exeter.ac.uk

†These authors contributed equally to this work

Competing interest: The authors declare that no competing interests exist.

**Abstract** Transactive response DNA binding protein 43 kDa (TDP43) proteinopathy, characterized by the mislocalization and aggregation of TDP43, is a hallmark of several neurodegenerative diseases, including Amyotrophic Lateral Sclerosis (ALS). In this study, we describe the development of a new model of TDP43 proteinopathy using human induced pluripotent stem cell (iPSC)-derived neurons. Utilizing a genome engineering approach, we induced the mislocalization of endogenous TDP43 from the nucleus to the cytoplasm without mutating the TDP43 gene or using chemical stressors. Our model successfully recapitulates key early and late pathological features of TDP43 proteinopathy, including neuronal loss, reduced neurite complexity, and cytoplasmic accumulation and aggregation of TDP43. Concurrently, the loss of nuclear TDP43 leads to splicing defects, while its cytoplasmic gain adversely affects microRNA expression. Strikingly, our observations suggest that TDP43 is capable of sustaining its own mislocalization, thereby perpetuating and further aggravating the proteinopathy. This innovative model provides a valuable tool for the in-depth investigation of the consequences of TDP43 proteinopathy. It offers a clinically relevant platform that will accelerate the identification of potential therapeutic targets for the treatment of TDP43-associated neurodegenerative diseases, including sporadic ALS.

## Introduction

ALS is a devastating neurodegenerative disorder characterized by the relentless degeneration of motor neurons (MNs) in both the brain and spinal cord (*Brown and Al-Chalabi, 2017*). This degeneration precipitates a cascade of symptoms including muscle weakness, atrophy, and paralysis, eventually leading to respiratory failure and death, typically within three to five years after the onset of symptoms (*Hardiman et al., 2017*).

A hallmark of ALS pathology is the aberrant behavior of the TAR DNA-binding protein 43 (hereafter TDP43). In ALS patients, TDP43, which normally resides in the nucleus, becomes mislocalized, forming

aggregates in the cytoplasm of neurons and glial cells (*Neumann et al., 2006*). This phenomenon, termed TDP43 proteinopathy, is implicated in the majority of ALS cases and is considered a central player in the disease's pathogenesis (*Ling et al., 2013*). TDP43 proteinopathy is a common feature in multiple age-associated neurodegenerative diseases, including ALS, limbic-predominant age-related TDP43 encephalopathy (LATE), frontotemporal dementia (FTD), and Alzheimer's disease (AD) (*de Boer et al., 2020*). TDP43 (encoded by *TARDBP*) is an RNA-binding protein that plays a critical role in RNA metabolism, encompassing RNA splicing (*Donde et al., 2019*; *Polymenidou et al., 2011*; *Arnold, 2012*; *Ling et al., 2015*), stabilization (*Sidibé et al., 2021*), and transport (*Nagano et al., 2020*; *Chu et al., 2019*), thus ensuring proper neuronal function (*Gimenez et al., 2023*). For example, recent studies have demonstrated cryptic exon (CE) inclusion triggered by loss of nuclear TDP43 (*Ling et al., 2015*), especially in transcripts of important neuronal genes *UNC13A* (*Brown et al., 2022*; *Ma et al., 2022*) and *STMN2* (*Klim et al., 2019*; *Melamed et al., 2019*). Additionally, TDP43 associates with the microRNA biogenesis machinery and affects microRNA expression (*Buratti et al., 2010*; *Kawahara and Mieda-Sato, 2012*; *Paez-Colasante et al., 2020*). Accordingly, widespread dysregulation of microRNA levels has been reported in spinal tissue obtained post-mortem from sporadic ALS cases (*Reichenstein et al., 2019*; *Emde et al., 2015*; *Figueroa-Romero et al., 2016*; *Gagliardi et al., 2019*).

Current cellular models aimed at investigating TDP43 mislocalization involve mutating the TDP43 nuclear-localization signal, using mutant versions identified in familial ALS patients, or the use of pharmacological agents to induce stress (*Barmada et al., 2010*; *Zuo et al., 2021*; *Ziff et al., 2023*; *Walker et al., 2015*). Despite the insights gained from these models, they harbor inherent limitations. Overexpression may not represent physiological conditions, TDP43 is found to be mutated in <0.5% of all ALS cases, and pharmacological induction might induce events unrelated to the disease. Furthermore, CEs identified in ALS/FTD cases are poorly conserved beyond primates, making it challenging to investigate the contribution of splicing defects to ALS pathophysiology using animal models (*Baughn et al., 2023*). Human induced pluripotent technology (iPSCs) offers a powerful platform that addresses some of these issues with animal models, enabling the development of human models of ALS in vitro (*Giacomelli et al., 2022*). A recent study employed sporadic ALS (spALS) spinal cord extracts to induce TDP43 proteinopathy in iPSC-derived MNs (*Smethurst et al., 2020*). However, the model's utility is limited by the small percentage of neurons showing TDP43 pathology and the difficulty in producing spALS spinal extracts consistently and in large quantities.

In this study, we present an innovative human model of TDP43 proteinopathy that allows us to trigger cytoplasmic mislocalization of endogenous non-mutated TDP43 on demand in healthy control iPSC-MNs at scale (*Figure 1A*). This technological advancement offers an unprecedented level of insight into TDP43 proteinopathy and its role in ALS.

## Results and discussion

We fused green fluorescent protein (GFP) to the C-terminus of *TARDBP* in its endogenous locus in healthy control iPSCs using CRISPR-Cas9 genome editing (*Figure 1A*) and selected iPSC clones with a homozygous knock-in (*Figure 1—figure supplement 1A*). We initially ascertained that the integration of GFP did not adversely impact the differentiation of iPSCs into motor neurons (MNs). To this end, iPSCs engineered with TDP43-GFP were subjected to differentiation directed towards the MN lineage, employing protocols we had established in previous studies (*Namboori et al., 2021*; *Hawkins et al., 2022*). Based on the expression of established MN markers - ISLET1 (ISL1) and neurofilament-M (NF-M) - we confirmed that the edited iPSCs (with GFP integration) efficiently differentiated into MNs (*Figure 1—figure supplement 1B*). Moreover, the TDP43-GFP knock-in MNs did not show significantly altered expression levels of *TARDBP*, *UNC13A*, and *STMN2*, nor did it cause any upregulation of cryptic transcripts in *STMN2* or *UNC13A* (*Figure 1—figure supplement 1C*). Overall, these results support the premise that the GFP knock-in had no detrimental effects on the iPSCs' ability to differentiate into MNs or TDP43 function.

These engineered iPSCs, hereafter called TDP43-GFP iPSCs, were differentiated into MNs (called TDP43-GFP MNs) where ~80% of the cells in culture immunostained positive for ISL1. We expressed an anti-GFP nanobody (12 kDa) using adeno-associated viruses (AAVs) in the TDP43-GFP MNs. Expression of the nanobody (deemed the control nanobody) did not affect TDP43 localization from the nucleus (*Figure 1B*). We engineered the nanobody with a nuclear export signal (NES). Expression

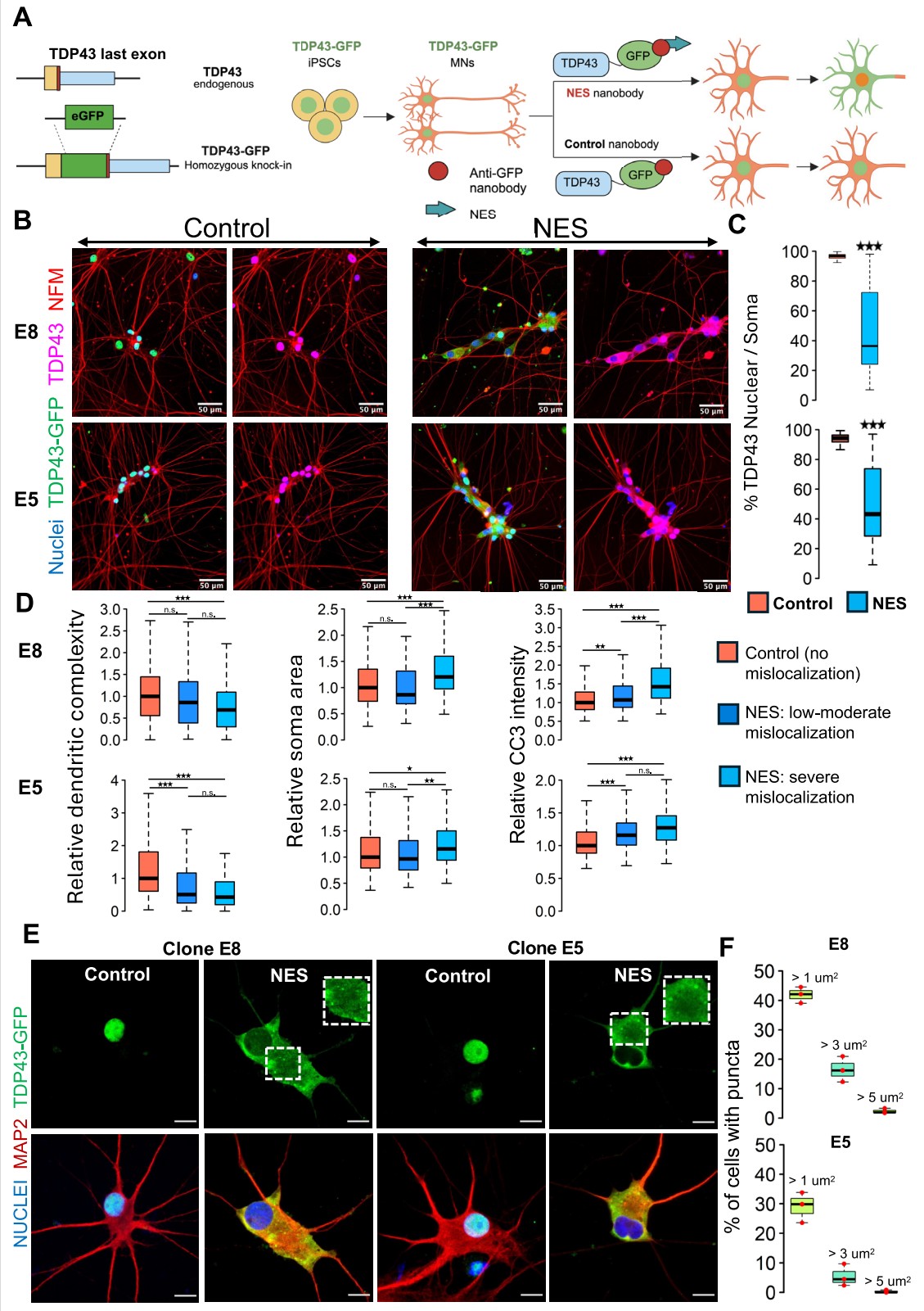

**Figure 1.** A human induced pluripotent stem cell (iPSC)-based model of TDP43 proteinopathy in Mns. (**A**) Schematic depicting genome editing of healthy iPSCs to knock-in GFP into the C-terminus of endogenous *TARDBP*, encoding TDP43. Yellow rectangle indicates the TDP43 last exon. Red vertical line indicates the STOP codon. Blue rectangle indicates the TDP43 3'UTR. The resulting TDP43-GFP iPSCs are differentiated into motor neurons (MNs) and transduced with adeno-associated viruses (AAVs) encoding anti-GFP nanobodies. The 'nuclear export signal (NES)' nanobody includes a

*Figure 1 continued on next page*

*Figure 1 continued*

sequence for a strong nuclear export signal, which transports nuclear TDP43-GFP into the cytoplasm. The control nanobody (lacking NES) has no effect on TDP43-GFP localization. Panel created in BioRender. (**B**) Representative images depicting TDP43 localization in two homozygous TDP43-GFP lines (**E8, E5**). MNs were transduced at day 20 of the differentiation process and fixed at day 40 for the immunostaining. TDP43 is expressed in the nucleus in the presence of the control nanobody and relocates to the cytoplasm in the presence of the NES nanobody. TDP43-GFP indicates signal from the anti-GFP antibody while TDP43 indicates signal from the anti-TDP43 antibody. Neurons were also stained with neurofilament-M (red) and Hoechst 33342 (blue nuclear stain). Scale bar = 50 µm. (**C**) Quantification of the % of nuclear TDP43 intensity over total TDP43 intensity in the nucleus + soma in individual neurons transduced with control or NES nanobodies. (**D**) Quantification of morphological defects and cleaved caspase 3 (CC3) levels in NES nanobody-treated MNs versus control. Mislocalized TDP43 (NES) causes a reduction in dendritic complexity (**D**), soma swelling, and elevation of CC3 levels, compared to neurons expressing nuclear TDP43 (control). Low-to-moderate mislocalization indicates neurons with >40% nuclear TDP43. Severe mislocalization indicates neurons with <40% nuclear TDP43. Measurements were normalized to data from the control nanobody condition. Panels C and D display data for the two homozygous TDP43-GFP lines (**E8 and E5**), transduced with nanobodies at day 20 and fixed for staining at day 40. N=3 independent differentiations per clone. At least 100 neurons were included per condition. (**E**) Representative images showing cytoplasmic TDP43 puncta at day 40 in homozygous TDP43-GFP MNs transduced with the NES nanobody at day 18. Scale bar = 15 µm. Images were captured with the Zeiss LSM880 Airyscan. (**F**) Quantification of the percentage of NES-expressing neurons displaying TDP43 puncta. The largest punctum per neuron was used for the analysis. Upper panel shows data for E8 neurons, while lower panel displays data for E5 neurons. N=3 independent differentiations per clone. At least 100 neurons were included per clone for the analysis. * indicates p<0.01, ** indicates p<0.001, *** indicates p<0.0001.

The online version of this article includes the following source data, source code, and figure supplement(s) for figure 1:

**Source code 1.** R script to analyse CC3 intensities for clone E5 motor neuron (MN).

**Source code 2.** R script to analyse CC3 intensities for clone E8 motor neuron (MN).

**Source code 3.** R script to analyse dendritic complexity for clone E5 motor neuron (MN).

**Source code 4.** R script to analyse dendritic complexity for clone E8 motor neuron (MN).

**Figure supplement 1.** Validation of the TDP43-GFP iPSCs.

**Figure supplement 1—source data 1.** screen for TDP43-GFP KI clones shown in *Figure 1—figure supplement 1A*.

**Figure supplement 1—source data 2.** Original gel files for PCR screen shown in *Figure 1—figure supplement 1A*.

**Figure supplement 2.** Phenotypic validation of the model.

**Figure supplement 2—source data 1.** CC3 intensities per soma for clone E5 MN.

**Figure supplement 2—source data 2.** Neurite complexity per soma for clone E5 MN.

**Figure supplement 2—source data 3.** CC3 intensities per soma for clone E8 MN.

**Figure supplement 2—source data 4.** Neurite complexity per soma for clone E5 MN.

of the NES-nanobody in the TDP43-GFP neurons caused relocation of the fusion protein to the cytoplasm with a concomitant loss in the nucleus (***Figure 1B,C***, ***Figure 1—figure supplement 2A***). We observed that TDP43 mislocalization leads to reduced dendrite complexity and neuronal soma swelling (***Figure 1D***, ***Figure 1—figure supplement 2A***) and an increase in cleaved caspase-3 activation (***Figure 1D***, ***Figure 1—figure supplement 2B***), which is an indicator of apoptosis. We confirmed that expression of the nanobodies in unedited iPSCs did not lead to apoptosis activation or dendrite defects, confirming that the observed phenotypes are due to TDP43 mislocalization (***Figure 1—figure supplement 2C***). Additionally, we observed TDP43 puncta reminiscent of aggregates in the cytoplasm of MNs displaying mislocalized TDP43 that varied in size and number across individual neurons (***Figure 1E,F***, ***Figure 1—figure supplement 1D***).

Having demonstrated that our model can recapitulate cellular features observed in sporadic ALS, we next explored the molecular consequences triggered by TDP43 proteinopathy. Previous studies have highlighted the prevalence of alternative splicing defects affecting transcripts expressed from *UNC13A* and *STMN2* in ALS (***Brown et al., 2022***; ***Ma et al., 2022***; ***Klim et al., 2019***; ***Melamed et al., 2019***). These splicing defects are considered hallmarks in the progression of the disease. Given the significance of this phenomenon, we sought to ascertain if MNs expressing the NES nanobody exhibit these characteristic splicing defects. We performed RT-qPCR analysis on motor neurons that expressed the control or NES nanobody. The results strikingly mirrored the commonly observed ALS profile, showing inclusion of cryptic exons in both *UNC13A* and *STMN2* and a reduction in the abundance of canonical transcripts (***Figure 1—figure supplement 2D***). This demonstrated that our model proficiently recapitulates the critical molecular features observed in sporadic ALS.

Next, we ventured to explore the molecular consequences triggered by TDP43 proteinopathy across the whole transcriptome. We performed transcriptomic analysis on day 40 TDP43-GFP MNs

to assess changes in the gene expression profiles due to TDP43 mislocalization. Principal component analysis (PCA) demonstrated that the primary axis of variance (90%) is related to the expression of the control and NES nanobody, while the much smaller second PC component is related to the two clones (*Figure 2A*). Subsequently, we conducted a differential gene expression analysis using DESeq2 and visualized the results via a volcano plot. Our analysis revealed differential expression of hundreds of genes with a fold change of ≥2 and false discovery rate (FDR)<0.01, suggesting a profound impact of TDP43 dysfunction on the transcriptome of MNs (*Figure 2B*). Gene ontology (GO) analysis of the differentially expressed genes highlighted a significant enrichment of pathways related to synaptic dysfunction and cytoskeletal defects in axons and dendrites amongst downregulated genes (*Figure 2C*). This finding is of considerable relevance to ALS pathogenesis, as neurons affected by ALS often exhibit impairments in these functions. In contrast, upregulated genes were enriched for GO terms related to RNA processing, including the nonsense-mediated decay (*Figure 2C*).

Given that alternative splicing (AS) defects are a hallmark of sporadic ALS, we utilized Leafcutter (*Li, 2018*) to analyze potential alternative splicing in MNs caused by TDP43 mislocalization. Our analysis identified alterations in the splicing of 175 genes, including *UNC13A* and *STMN2* (ΔPSI >0.1, adjusted p-value <0.01). To gain functional insights into these splicing defects, we performed pathway enrichment analysis on affected genes, identifying a significant enrichment of terms related to synaptic development (FDR <0.01). Furthermore, to gain a mechanistic understanding of the role of TDP43 in the observed splicing defects, we compared the splicing results with TDP43 eCLIP data generated in the SH-SY5Y neuroblastoma cell line (*Tam et al., 2019*). Interestingly, only 35% of the splicing changes were proximal to a TDP43 binding site (*Figure 2—figure supplements 1 and 2*). This suggests that the loss of TDP43 from the nucleus may be responsible for a subset of the observed splicing defects, but not all. This indicates the possibility of additional indirect underlying mechanisms that may include dysregulation of other RBPs or epitranscriptomic modifications of the RNA targets (*McMillan et al., 2023*). To gain deeper insights into the observed splicing patterns due to TDP43 mislocalization, we employed the Oxford Nanopore Technologies (ONT) long-read sequencing platform to generate RNA-seq data at the isoform level. Our investigation predominantly focused on *STMN2* due to its significance in ALS and a substantial number of reads (>100) mapping to this gene. Employing our FICLE pipeline (*Leung et al., 2023*), we identified a total of 476 isoforms related to *STMN2*. Out of these, 17 isoforms were congruent with the exonic structure of known reference isoforms. Notably, 40 isoforms, accounting for 8.4% of the total, were characterized by a cryptic exon (CE) starting at chr8:80529057 (*Figure 2—figure supplement 3A*). These isoforms with the CE were exclusively expressed in TDP43-GFP MNs that expressed the NES nanobody. Furthermore, the ONT data revealed that *STMN2* isoforms exhibit variable CE lengths, with different CE lengths corresponding to widely varying expression levels of the parent isoform (*Figure 2—figure supplement 3B*). Importantly, the isoforms containing the CE were predominantly short, truncated, and predicted to be non-protein coding. Surprisingly, four of these CE-containing isoforms manifested a novel exon, 114 bp in length, positioned upstream of the CE (*Figure 2—figure supplement 3A*). Our findings reveal a potentially significant variability in the splicing alterations induced by TDP43 proteinopathies, even within a single gene. This highlights the power of using long-read sequencing as a method for uncovering nuanced changes in splicing alterations in neurodegenerative diseases.

We compared our splicing results with publicly available transcriptomic data generated from cortical neuronal nuclei with or without TDP43 purified from ALS patient tissue post-mortem. At a stringent FDR threshold of 0.01 and ΔPSI >0.1, we detected 90 genes as alternatively spliced in nuclei that showed a loss of TDP43 using Leafcutter. Thirty-four out of these 90 genes were also detected in our iPSC model (*Figure 2D*), indicating strong concordance between the iPSC model and patient data, despite the differences in sample origin and neuronal subtype. Importantly, all 34 genes displayed an identical splicing event. To evaluate whether the observed splicing changes were due to nuclear loss of TDP43, we compared our splicing results with transcriptomic data generated in iPSC-derived MNs and iNeurons after TDP43 knockdown (KD) (*Figure 2E and F*). In accordance with the model that nuclear loss of TDP43 drives splicing changes, we observed a significant overlap between splicing defects due to a global loss of TDP43 and our mislocalization model (*Figure 2E and F*). However, a number of genes that displayed splicing changes due to TDP43 mislocalization were not affected by TDP43 KD in either dataset (143 genes for the GSE121569 and 79 genes for the PRJEB42763 dataset). This indicates that nuclear loss of TDP43 alone cannot entirely explain the widespread defects in AS.

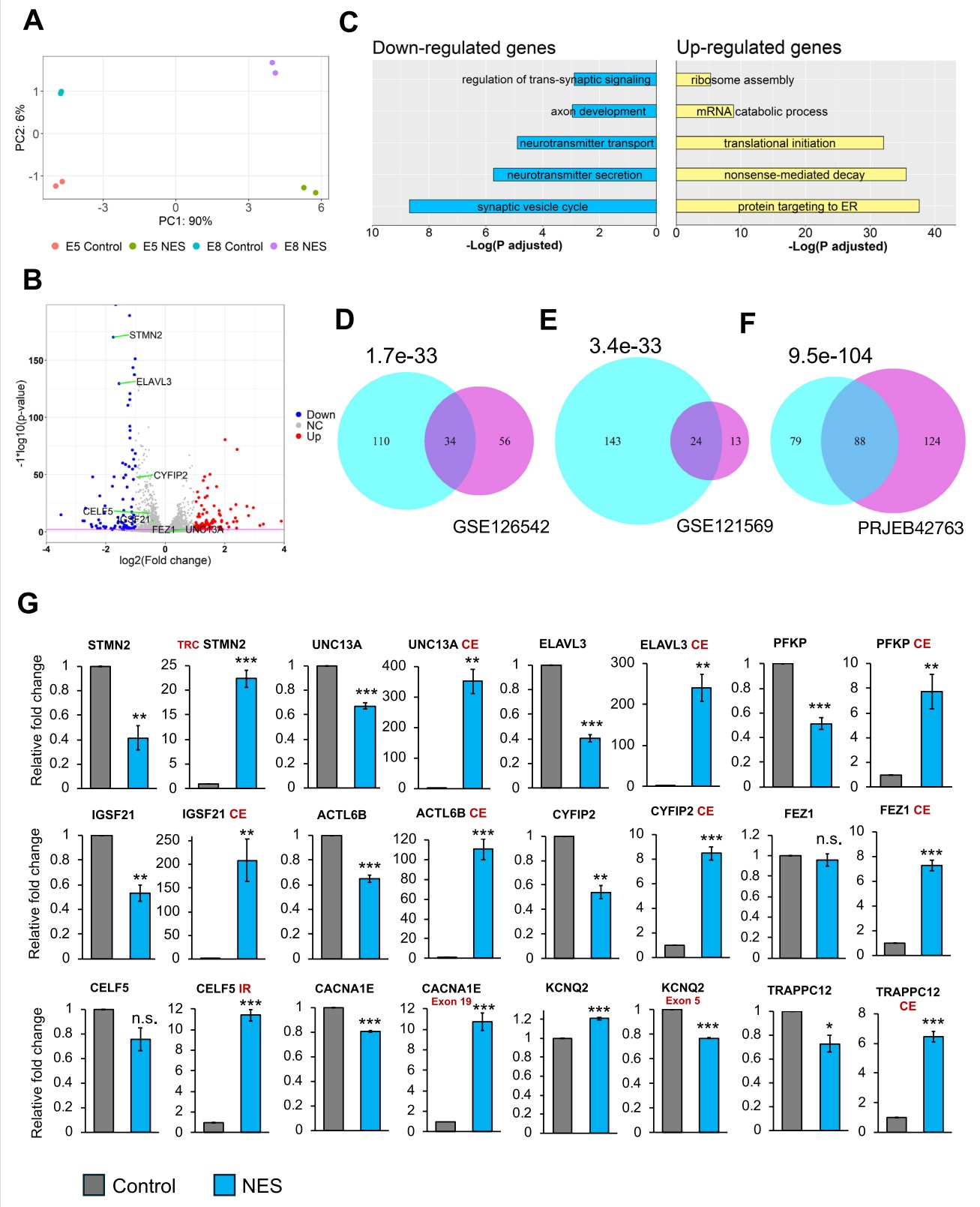

**Figure 2.** Transcriptional consequences of TDP43 mislocalization. (**A**) Principal component analysis of gene counts for E8 and E5 iPSC-MNs. (**B**) Volcano plot displaying genes differentially expressed due to TDP43 mislocalization. Red: upregulated genes, Blue: downregulated genes, NC: No Change. Horizontal line denotes a p-value threshold of 0.01. (**C**) Gene ontology enrichment analysis on the differentially expressed genes in the TDP43 model. The top five enriched pathways in upregulated and downregulated genes have been displayed. (**D**, **E**, **F**) Venn diagram displaying the overlap

*Figure 2 continued on next page*

*Figure 2 continued*

between mis-spliced genes identified due to TDP43 mislocalization in the TDP43-GFP induced pluripotent stem cell (iPSC) MNs (cyan circles) with publicly available transcriptomic datasets; D: Cortical neuronal nuclei displaying TDP43 depletion obtained from Amyotrophic Lateral Sclerosis (ALS)/ frontotemporal dementia (FTD) patient tissue (GSE126542), E: TDP43 knockdown using siRNAs in healthy iPSC-MNs (GSE121569), F: TDP43 knockdown using CRISPRi in healthy iPSC-iNs (PRJEB42763). P-values were estimated using a hypergeometric distribution. Only genes with detectable expression in both datasets were used for splicing analysis. (**G**) RT-qPCR validating alternative splicing changes resulting from TDP43-GFP mislocalization in iPSC-MNs. MN samples were lysed at day 30, 12 days post-transduction with AAVs. Replicates were three independent differentiations of homozygous TDP43-GFP knock-in lines, (two of E5, one of E8). CE = cryptic exon, TRC = truncated, IR = intron retention. *CACNA1E* and *KCNQ2* displayed alternate exon usage. The exons that were included in the NES samples have been indicated. ** indicates p<0.01. *** indicates p<0.001.

The online version of this article includes the following source data, source code, and figure supplement(s) for figure 2:

**Source data 1.** DESeq2 output related to *Figure 2*.

**Source data 2.** Leafcutter analysis to analyse splicing changes due to TDP43 mislocalisation.

**Source code 1.** R script to analyse differential gene expression data for *Figure 2*.

**Figure supplement 1.** UCSC Genome browser screen shots depicting cryptic exon inclusion (*UNC13A, STMN2, ELAVL3*) and intron retention (*CELF5*) resulting from TDP43 mislocalization.

**Figure supplement 2.** UCSC Genome browser shots depicting cryptic exon inclusion in *CYFIP2, FEZ1,* and *IGSF21* resulting from TDP43 mislocalization.

**Figure supplement 3.** Quantification of variation in the *STMN2* CE.

**Figure supplement 4.** RT-qPCR showing alternative splicing errors in unedited induced pluripotent stem cell (iPSC) motor neurons (MNs), 10 days post-addition of shRNAs targeting TDP43, or controls.

---

To generate a robust list of AS events associated with TDP43 pathology, we compared genes identified in our RNA-seq data with those from ALS/FTD post-mortem tissue. We applied an adjusted p-value threshold of 0.01 and a ΔPSI >0.1. Additionally, to ensure further confidence in the identified changes, we required that each event be statistically significant at a stringent p-value threshold of 1e-4 in at least one dataset. Our analysis identified 12 genes, including *STMN2* and *UNC13A*, that we further investigated using RT-qPCR. This confirmed AS events in all the genes identified, where 9/12 genes displayed a decrease in expression of their canonical transcript levels (*Figure 2G*). Furthermore, most of these genes also displayed AS changes due to TDP43 KD in iPSC-MNs, although the extent of these changes for a subset of the genes tested was less dramatic (*Figure 2—figure supplement 4*).

A key drawback with earlier TDP43 models was the inability to characterize early changes after TDP43 mislocalization in neurons. To enable inducible control of TDP43 cytoplasmic localization in our model, we expressed the nanobody under a doxycycline-inducible promoter, and the entire circuit was knocked into the *AAVS1* locus in the E8 homozygous TDP43-GFP iPSCs. We called these iPSCs TDP43-GFP-CTRL or TDP43-GFP-NES iPSCs (*Figure 3A*). We first confirmed that we could induce TDP43 mislocalization in the TDP43-GFP-CTRL/NES iPSC-derived MNs using doxycycline (Dox). Dox (1 μg/ml) triggered significant TDP43 mislocalization in the TDP43-GFP-NES MNs, while the TDP43-GFP-CTRL MNs and the TDP43-GFP-NES MNs without Dox maintained nuclear TDP43 (*Figure 3B*). Furthermore, we observed cytoplasmic TDP43 puncta in the NES line (*Figure 3C*), and a significant increase in phosphorylated TDP43 without a change in the total TDP43 protein levels upon mislocalization (*Figure 3D, E*, *Figure 3—figure supplement 2*). We noted that the TDP43 mislocalization alone did not induce significant levels of G3BP1+ stress granules (SG) (*Figure 3—figure supplement 1*). When treated with sodium arsenite, both control and NES MNs displayed cytoplasmic SG (*Figure 3—figure supplement 1*). However, in NES MNs treated with SA, only a subset of cytoplasmic TDP43 puncta co-localized with G3BP1+ stress granules (*Figure 3—figure supplement 1*).

To detect early changes in expression and splicing post-TDP43 mislocalization, we performed a time-course analysis of TDP43-GFP-NES at 4-, 8-, 12-, and 24-hr post-Dox addition. Notably, we observed TDP43 mislocalization as early as 8 hr after Dox treatment (*Figure 4A*), which was accompanied by significant AS errors in all 12 genes (*Figure 4B*). This suggests that dysfunctional AS could be one of the incipient molecular events in ALS pathogenesis due to TDP43 mislocalization. Transcriptomic analysis of day 40 MNs uncovered splicing defects, including cryptic exon inclusions and isoform switching in 494 genes (adjusted p-value <0.01 and ΔPSI >0.1), with significant enrichment in pathways related to synaptic function and the cytoskeleton (*Figure 3—figure supplement 3*).

A significant area of research in the field of ALS involves pinpointing the upstream triggers responsible for causing the mislocalization of TDP43. The underlying premise of these investigations is that

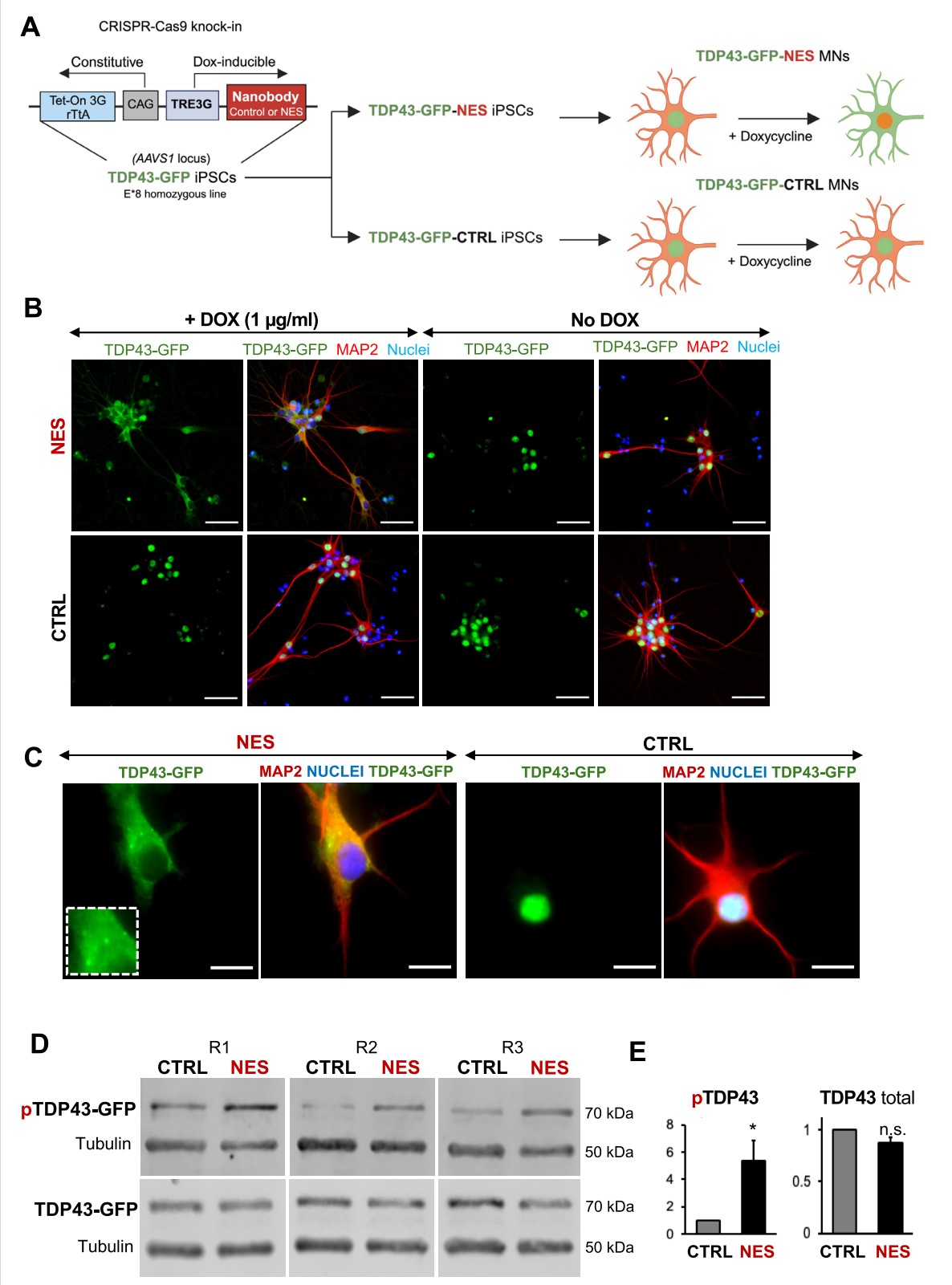

**Figure 3.** An inducible model of TDP43 mislocalization. (**A**) Schematic depicting the knock-in of the nanobody (control or nuclear export signal , NES) into the human *AAVS1* safe harbour locus of our E8 homozygous TDP43-GFP cell line. Nanobody expression was under the control of a doxycycline (Dox)-inducible promoter. Addition of Dox is expected to induce TDP43 mislocalization in the TDP43-GFP-NES motor neurons (MNs) but not in the TDP43-GFP-Control (TDP43-GFP-CTRL) MNs. Panel created in BioRender. (**B**) Immunofluorescent staining showing TDP43 localization in our TDP43-

*Figure 3 continued on next page*

*Figure 3 continued*

GFP-CTRL or TDP43-GFP-NES line in response to Dox. Mislocalization only occurs in the TDP43-GFP-NES line with Dox treatment, while all other conditions maintain nuclear TDP43. MNs were fixed and stained at day 35, 15 days post-Dox addition. Scale bar = 50 µm. (**C**) TDP43 localization in the TDP43-GFP-NES or –CTRL cell lines at day 40, 20 days post-Dox addition. The box highlights cytoplasmic TDP43 puncta in the NES line. Scale bar = 20 µm. (**D**) Western blots of total and phosphorylated TDP43 in TDP43-GFP-NES or –CTRL cell lines at day 40, 20 days post-Dox addition. Alpha-tubulin was used as a loading control. (**E**) Quantification of total and phosphorylated TDP43-GFP from Figure D. Total TDP43-GFP remains stable, while there is a significant increase in phosphorylated TDP43-GFP in the NES lines. Each sample was normalized to alpha-tubulin, pTDP43 samples were also normalized to total TDP43 levels. Replicates are three independent differentiations of TDP43-GFP-NES or –CTRL. pTDP43-GFP=phosphorylated TDP43-GFP. * indicates p<0.05. Error bars indicate SEM.

The online version of this article includes the following source data, source code, and figure supplement(s) for figure 3:

**Source data 1.** PDF of labelled uncropped western blots shown in *Figure 3D*.

**Source data 2.** unedited original files for western blots shown in *Figure 3D*.

**Figure supplement 1.** Representative images of TDP43-GFP-NES or –CRTL lines with and without SA exposure (500 µM for 60 min), 3 days in Dox (1 µg/mL).

**Figure supplement 2.** Uncropped western blot from *Figure 3D*.

**Figure supplement 3.** RNA-seq analysis of MNs displaying TDP43 mislocalisation at day 40.

**Figure supplement 3—source code 1.** R script to analyse differential gene expression data for *Figure 3—figure supplement 3*.

**Figure supplement 3—source code 2.** R script to analyse splicing changes for *Figure 3—figure supplement 3*.

**Figure supplement 3—source data 1.** Leafcutter analysis to analyse splicing changes due to TDP43 mislocalisation related to *Figure 3—figure supplement 3*.

**Figure supplement 3—source data 2.** Leafcutter analysis to analyse splicing changes due to TDP43 mislocalisation related to *Figure 3—figure supplement 3*.

**Figure supplement 3—source data 3.** DESeq2 output related to *Figure 3—figure supplement 3*.

**Figure supplement 3—source data 4.** Sample details related to RNA-seq analysis shown in *Figure 3—figure supplement 3*.

**Figure supplement 4.** Differential expression analysis of microRNAs.

**Figure supplement 4—source data 1.** Raw counts for microRNA expression analysis comparing healthy MN to those displaying TDP43 mislocalization.

**Figure supplement 4—source data 2.** DESeq2 output for differential microRNA analysis comparing healthy MN to those displaying TDP43 mislocalization.

**Figure supplement 4—source data 3.** Raw counts for microRNA expression analysis comparing healthy MN to those displaying TDP43 knockdown.

**Figure supplement 4—source data 4.** DESEq2 output for microRNA expression analysis comparing healthy MN to those displaying TDP43 knockdown.

**Figure supplement 4—source data 5.** Sample details related to RNA-seq analysis shown in *Figure 3—figure supplement 4*.

---

eliminating the trigger may potentially reverse the mislocalization of TDP43. However, it remains to be confirmed whether this hypothesis holds true. We wanted to ascertain whether TDP43, once mislocalized, self-perpetuates in its mislocalized state even after the initial trigger has been removed.

To investigate whether TDP43 mislocalization is self-perpetuating, we tagged both the control and NES nanobody with a V5 tag to track their expression. These were introduced into TDP43-GFP MNs using lentiviruses under a doxycycline (Dox)-inducible promoter. We induced TDP43 mislocalization by adding Dox to iPSC-derived MNs at day 20. After five days (day 25), Dox was withdrawn, and neurons were harvested 21 days later (day 46). As a benchmark of successful mislocalization, cultures were continuously treated with Dox (Constant Dox), while neurons expressing the control nanobody served as controls, displaying nuclear TDP43. TDP43 localization was assessed using immunofluorescence microscopy, and nanobody expression was evaluated via V5 immunostaining.

As expected, neurons in the Constant Dox NES condition exhibited significant TDP43 mislocalization, while TDP43 remained nuclear in the Constant Dox control MNs, confirming the specificity of Dox-induced mislocalization (*Figure 5A and B*). Following Dox withdrawal, TDP43 localization was largely restored to the nucleus. However, a subset of neurons retained cytoplasmic TDP43 even after 21 days (*Figure 5A*). Super-resolution microscopy further revealed persistent cytoplasmic TDP43 in these neurons, despite near undetectable nanobody expression (*Figure 5C*), suggesting that in some cells, TDP43 mislocalization may become self-sustaining. Our data indicates that removal of the initial trigger may be insufficient to completely reverse TDP43 once it has been mislocalized.

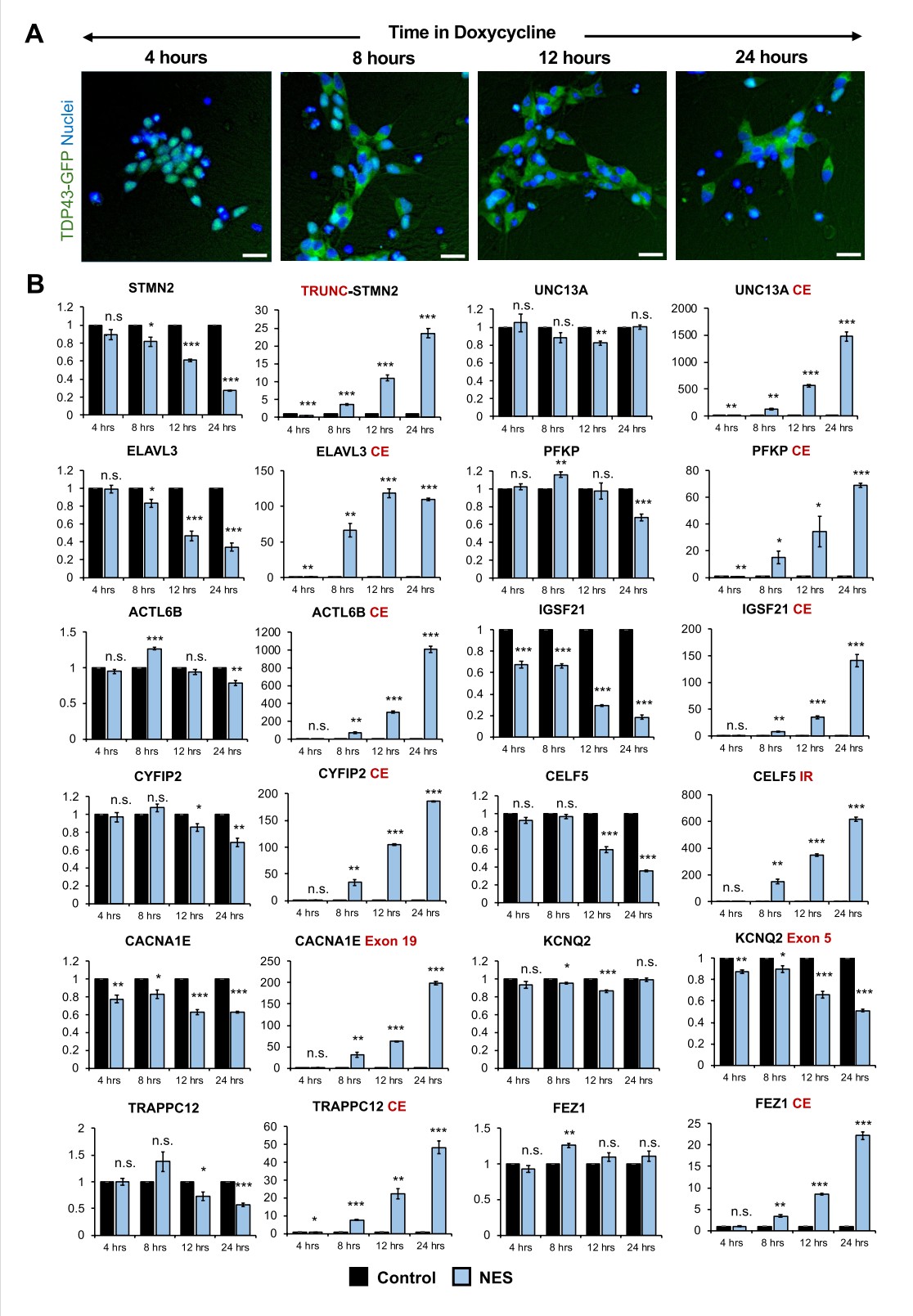

**Figure 4.** Splicing defects are one of the earliest changes downstream of TDP43 mislocalization. (**A**) Immunocytochemical staining of TDP43-GFP-NES MNs at 4, 8, 12, and 24 hr following 1 µg/ml doxycycline addition. TDP43 mislocalization is observed within 8 hr. Dox was added to motor neuron (MN) day 20. Scale bar = 25 µm. (**B**) RT-qPCR of alternative splicing changes at 4, 8, 12, and 24 hr following 1 µg/ml doxycycline addition. Significant cryptic transcript expression in all genes is detected at 8 hr. Error bars show SEM. N=3 independent differentiations of the TDP43-GFP-CTRL/-NES lines. CE

*Figure 4 continued on next page*

*Figure 4 continued*

= cryptic exon, TRC = truncated, IR = intron retention. *CACNA1E* and *KCNQ2* displayed alternate exon usage. The exons that were included in the nuclear export signal (NES) samples have been indicated. * indicates p<0.05, ** indicates p<0.01, *** indicates p<0.001.

Finally, since TDP43 is involved in microRNA biogenesis, we sought to analyze changes in the microRNA profiles associated with TDP43 mislocalization. For this purpose, we carried out small RNA sequencing on day 40 MNs after triggering TDP43 mislocalization at day 20 by the addition of doxy-cycline. Our analysis revealed a striking alteration in the landscape of microRNA expression as a result of TDP43 mislocalization. Principal component analysis (PCA) demonstrated distinct separation of the control and mislocalized samples (PC1 77%), emphasizing the profound impact of TDP43 mislocalization on microRNA profile (*Figure 3—figure supplement 4A*).

Around 150 microRNAs were found to be significantly altered (FDR <0.01 and fold change ≥1.5) upon TDP43 mislocalization. Our data captured downregulation of the microRNAs miR-218, and miR-9, which have previously been shown to be downregulated in ALS MNs (*Reichenstein et al., 2019*; *Zhang et al., 2013*). Interestingly, these changes were not unidirectional; almost equal numbers of microRNAs were upregulated or downregulated (*Figure 3—figure supplement 4B*). The results of this study indicate that TDP43 mislocalization leads to global dysregulation of microRNA expression in iPSC-derived MNs. However, it is noteworthy that our observations contrast with those found in postmortem tissue studies, where all differentially expressed microRNAs were reported to be down-regulated. This divergence in findings may suggest that the uniform downregulation observed in postmortem tissues could be attributed to changes that are triggered by end-stage dying neurons, which might not represent the whole spectrum of molecular alterations, especially early in the disease progression.

We wanted to evaluate whether alterations in microRNA expression profiles induced by TDP43 mislocalization are congruent with the changes caused by TDP43 knockdown. For this investigation, small RNA sequencing was utilized to assess microRNA expression in motor neurons (MNs) at day 30 following the knockdown of TDP43 using shRNAs. Again, principal component analysis revealed a distinct separation between all control and TDP43 knockdown samples, indicating a clear impact of TDP43 knockdown on microRNA expression profiles (*Figure 3—figure supplement 4C*). However, a striking contrast was observed in the number of microRNAs affected by TDP43 knock-down as compared to TDP43 mislocalization. Specifically, in the case of TDP43 knockdown, only one microRNA (miR-1249–3 p) exhibited a significant change (FDR <0.01 and a fold change of 1.5) (*Figure 3—figure supplement 4D*). This is in stark contrast to the observations made when TDP43 was mislocalized, where over 150 microRNAs displayed altered expression. To avoid setting thresh-olds in our comparisons, we first ranked the microRNAs impacted by TDP43 knockdown based on their fold changes. Subsequently, we utilized gene set enrichment analysis (GSEA) to determine if the microRNAs disrupted by TDP43 mislocalization significantly coincided with our knockdown data. We observed a substantial overlap between microRNAs that were downregulated due to TDP43 mislo-calization and those downregulated following TDP43 knockdown (*Figure 3—figure supplement 4E*). However, we did not observe a significant overlap for microRNAs that were upregulated in both scenarios (*Figure 3—figure supplement 4F*). Our data suggest that mislocalized TDP43 selectively affects the expression of a subset of microRNAs. Additionally, these observations highlight the possi-bility that the nuclear loss and cytoplasmic gain of TDP43 induce distinct molecular alterations within motor neurons that converge to hasten neuronal dysfunction and demise.

However, we acknowledge that another explanation for the observed differences between microRNA dysregulation and splicing could be the extent of nuclear TDP43 loss. Our knockdown approach resulted in a 60% reduction in TDP43 transcripts. Though this was sufficient to cause splicing defects, it is possible that microRNA dysregulation might be more resilient to TDP43 loss as compared to splicing changes.

## Limitations of the study

We could not confirm whether the observed cytoplasmic puncta are true aggregates. Attempts to immunostain neurons for phosphorylated TDP43 were unsuccessful. Using the Cosmo Bio antibody (TIP-PTD-M01A), we detected no signal, while the Proteintech antibody (22309–1-AP) showed nuclear phosphorylation even in control neurons. To address this, we performed western blot analysis with

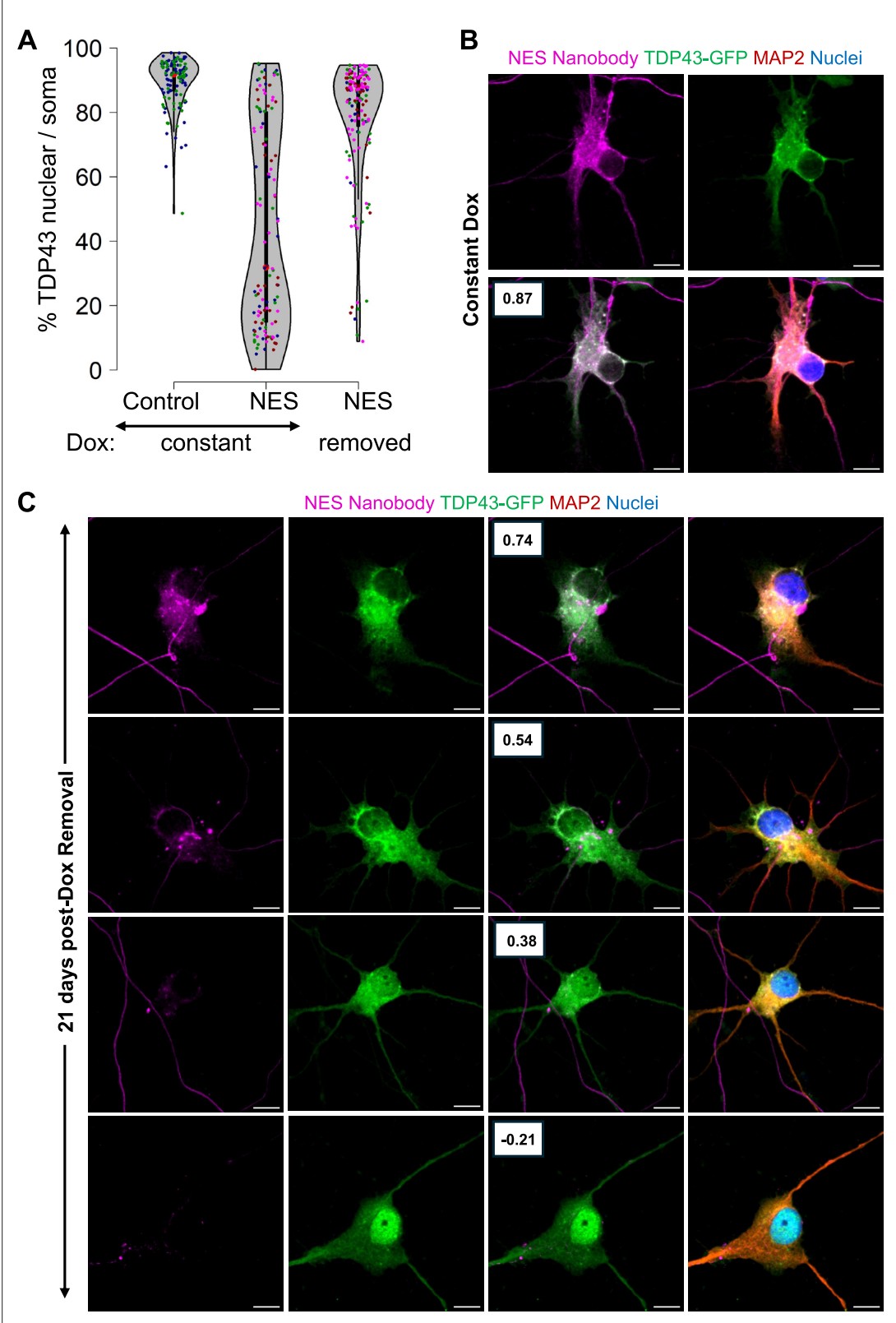

**Figure 5.** Persistent mislocalization of TDP43 in neurons post-Dox withdrawal. (**A**) TDP43 localization in the E8 TDP43-GFP motor neurons (MNs) expressing Dox-inducible V5-tagged nanobodies. Control: control-V5 nanobody. NES: NES-V5 nanobody. Constant: neurons were treated with Dox continuously. Removed: Dox was withdrawn five days after its addition. N=2 for the constant +control condition. N=4 for the nuclear export signal (NES) conditions. Each replicate is indicated by a coloured dot. (**B, C**) Representative images of the TDP43-GFP MNs expressing NES-V5 nanobody

*Figure 5 continued on next page*

*Figure 5 continued*

under constant Dox treatment (**B**) and 21 days post-Dox withdrawal (**C**). Boxes indicate the Pearson correlation coefficient between the TDP43-GFP and V5-nanobody signals across individual pixels within the soma. A higher coefficient indicates higher co-localization. Scale bar = 10 μm. Images were captured with the Zeiss LSM880 Airyscan.

the Cosmo Bio antibody, which clearly detected hyperphosphorylated TDP43 in whole-cell lysates (*Figure 3D and E*). Given these results, we refer to these structures as 'puncta' rather than aggregates.

TDP43 undergoes alternative splicing within exon 6 in response to induced hyperexcitability in iPSC-derived iNeurons, generating truncated proteins. These isoforms are detected in lumbar motor neurons; however, their levels remained unchanged between ALS patients and healthy tissue (*Weskamp et al., 2020*). Since our GFP tag is positioned on the C-terminus, our system cannot manipulate these truncated fragments as the tag is lost in these isoforms. But these isoforms, if present, should be detectable using the Proteintech antibody against total TDP43, which recognizes N-terminal TDP43 epitopes. However, western blot analysis, even 20 days after inducing TDP43 mislocalization, showed no truncated fragments. This suggests that TDP43 mislocalization alone is insufficient to generate significant levels of truncated isoforms.

## Summary

In summary, our unique iPSC model of TDP43 proteinopathy offers an unprecedented access to investigate cellular and molecular events in human neurons in a temporal fashion downstream of TDP43 mislocalization. This model faithfully captures the neuronal loss and cytoplasmic accumulation characteristics of TDP43 proteinopathies. Beyond neurons, our model paves the way for inquiries into the roles of non-neuronal cells, such as oligodendrocytes and astrocytes, which also exhibit TDP43 proteinopathy, in contributing to neuronal dysfunction in a cell non-autonomous manner (*Barton et al., 2021*; *James et al., 2022*; *Smethurst et al., 2020*; *Licht-Murava et al., 2023*). We expect this model to not only enhance our understanding of ALS but also other TDP43 proteinopathies and accelerate efforts into developing therapies against these devastating neurodegenerative diseases.

# Materials and methods

**Key resources table**

| Reagent type (species) or resource | Designation | Source or reference | Identifiers | Additional information |
|---|---|---|---|---|
| Cell line (human iPSCs, female) | 'parent' Healthy adult donor iPSC line | Coriell Institute for Medical Research | GM23280A | |
| Cell line (human iPSCs, female) | TDP43-GFP (E5 or E8) | This paper | | GFP Knock-in iPSC lines generated from healthy 'parent' line. Maintained in A. Bhinge Lab. |
| Cell line (human iPSCs, female) | TDP43-GFP-NES or TDP43-GFP-CTRL | This paper | | Nanobody (CTRL or NES) *AAVS1* knock-in line generated from TDP43-GFP E8 line. Maintained in A. Bhinge Lab. |
| Cell line (human) | AAVpro 293T | Takara | #6322773 | For AAV generation |
| Cell line (human) | Lenti-X 293T | Takara | #632180 | For lentiviral generation |
| Antibody | Anti-GFP (goat polyclonal) | Novus/ Bio-Techne | #NB100-1770 | IF (1:1000) |
| Antibody | Anti-human MAP2 (chicken polyclonal) | GeneTex | #GTX82661 | IF (1:10,000) |
| antibody | Anti-human TDP-43 (rabbit polyclonal) | Proteintech | #10782–2-AP | IF (1:800) WB (1:1000) |
| Antibody | Anti-human Cleaved caspase-3 (rabbit monoclonal) | Cell Signalling | #9664 L | IF (1:400) |

*Continued on next page*

*Continued*

| Reagent type (species) or resource | Designation | Source or reference | Identifiers | Additional information |
|---|---|---|---|---|
| Antibody | Anti-human Neurofilament-M (mouse monoclonal) | Merck | #Mab1621 | IF (1:1000) |
| Antibody | Anti-human ISLET-1 (rabbit monoclonal) | Abcam | #Ab109517 | IF (1:500) |
| Antibody | Anti V5 tag (mouse monoclonal) | Invitrogen | #MA5-15253 | IF (1:1000) |
| Antibody | Anti-human G3BP1 (rabbit polyclonal) | Proteintech | #13057–2-AP | IF (1:1000) |
| Antibody | Anti-human phospho-TDP43 (mouse monoclonal) | Cosmo Bio | #TIP-PTD-M01A | WB (1:1000) |
| Antibody | Anti-human alpha-tubulin (mouse monoclonal) | Abcam | #AB7291 | WB (1:1000) |
| Antibody | Anti-human alpha-tubulin (rabbit polyclonal) | Abcam | #AB4074 | WB (1:1000) |
| Antibody | IRDye 680RD anti-mouse (goat polyclonal) | Li-Cor | #926–68070 | WB (1:5000) |
| Antibody | IRDye 800CW anti-rabbit (goat polyclonal) | Li-Cor | #926–32211 | WB (1:5000) |
| Recombinant DNA reagent | Addgene plasmid 52343 | Addgene. Su-Chun Zhang; *Qian et al., 2014* | | Donor plasmid for *AAVS1* nanobody knock-in |
| Recombinant DNA reagent | RRID:Addgene 126582 | Addgene. Michael Guertin *Sathyan et al., 2019*. | | sgRNA targeting the human *AAVS1* locus |
| Sequence-based reagent | Nanobody sequence | Addgene plasmid 136619; *Farrants et al., 2020* | Kai Johnsson; | |
| Sequence-based reagent | NES nanobody | This paper | NES nanobody sequence | MNLVDLQKKLEELELDEQQ |
| Sequence-based reagent | NES nanobody | This paper | NES nanobody sequence | IDEAAKELPDANA |
| Sequence-based reagent | TDP-43 C-terminus sgRNA | This paper | sgRNA | GAATGTAGACAGTGGGGTTG |
| Sequence-based reagent | shRNA targeting TARDBP | Broad Institute GPP portal | | GCAATAGACAGTTAGAAAGAA |
| Sequence-based reagent | Control shRNA | Broad Institute GPP portal | | TAGGAATTATAATGCTTATCTA |
| Commercial assay or kit | miRNeasy Micro Kit | Qiagen | #217084 | RNA extraction |
| Commercial assay or kit | High-capacity cDNA kit | Thermo Fisher | #4368814 | Reverse transcription reaction |
| Commercial assay or kit | GoTaq qPCR Master Mix | Promega | #A6001 | qPCR |
| Commercial assay or kit | NEB Monarch total RNA Miniprep kit | New England Biolabs | #T2010S | RNA extraction |
| Commercial assay or kit | NEBNext rRNA depletion kit | New England Biolabs | #E7405L | |

*Continued on next page*

*Continued*

| Reagent type (species) or resource | Designation | Source or reference | Identifiers | Additional information |
|---|---|---|---|---|
| Commercial assay or kit | NEBNext Ultra II RNA library kit | New England Biolabs | #E7770S | |
| Commercial assay or kit | AAVpro Extraction Solution (Takara) | Takara | #6235 | AAV Purification |
| Commercial assay or kit | Calcium phosphate transfection kit | Takara | #631312 | Transfection for AAV and lentiviral generation |
| Chemical compound, drug | Doxycycline | Merck | #324385 | (1 µg/mL) |
| Software, algorithm | Cell Profiler | Cell Profiler; *Stirling et al., 2021* | | |
| Other | Hoechst 33342 | Thermo Fisher Scientific | 62249 | IF (1:1000) |

## Genome editing

CRISPR-Cas9 GFP knock-in into the TDP43 C-terminus was performed on healthy iPSC lines (GM23280A, Coriell Institute for Medical Research). The iPSCs were maintained as colonies on Matrigel (Corning) in StemFlex (StemCell Technologies). The identity of the iPSCs was authenticated using immunostaining of OCT4 and NANOG and the ability to differentiate into neurons. Cells were routinely tested for mycoplasma contamination every three months. All tests were negative.

The eGFP donor plasmid was designed to add a 13 amino acid linker between the TDP43 second last codon and eGFP sequence as described previously (*Barmada et al., 2010*) and carried 400 bp homology arms on the 5' and 3' ends of the TDP43 stop codon. For genome editing, iPSCs were transfected with the eGFP donor plasmid and sgRNA targeting TDP43 C-terminus (GAATGTAGACAG TGGGGTTG).

Knock-in of GFP was confirmed via PCR and the sequence was validated using Sanger sequencing. Two independently edited homozygous clones ('E5' and 'E8') were selected. The nanobody sequence was obtained from Addgene plasmid 136619 (a kind gift from Kai Johnsson *Farrants et al., 2020*) and synthesized as a gBlock from IDT. The NES sequence from the MAPKK gene MNLVDLQK-KLEELELDEQQ was added onto the nanobody sequence to create the NES nanobody at its C-terminus using PCR primers. An unrelated sequence IDEAAKELPDANA was used to create the control nanobody.

For the TDP43-GFP-NES and TDP43-GFP-CTRL lines, the donor vector was created by cloning in the NES or control nanobody sequences into Addgene plasmid 52343 (a kind gift from Su-Chun Zhang *Qian et al., 2014*). Homozygous TDP43-GFP iPSCs derived from clone E8 were transfected with the donor vectors, and a sgRNA targeting the human *AAVS1* locus (Addgene 126582, a kind gift from Michael Guertin *Sathyan et al., 2019*). The nanobody sequence was PCR amplified from Addgene. Edited cells were selected with puromycin (0.5 µg/ml) treatment for a minimum of 4 days.

## Cell culture

Motor neurons were generated from iPSCs as described previously (*Namboori et al., 2021*; *Hawkins et al., 2022*). Briefly, iPSCs were plated onto Matrigel and differentiated by treatment with neuronal differentiation media (DMEM/F12: Neurobasal in a 1:1 ratio, HEPES 10 mM, N2 supplement 1%, B27 supplement 1%, L-glutamine 1%, ascorbic acid 5 µM) supplemented with SB431542 (40 µM), CHIR9921 (3 µM), and LDN98312 (0.2 µM) from day 0 until day 4. Cells were caudalized by treatment with 0.1 µM retinoic acid starting at day 2 and ventralized with 1 µM purmorphamine starting at day 4 and continued until day 10. At day 8, progenitors were replated onto poly-D-lysine/laminin-coated wells and differentiated with 10 µM DAPT for 3 days. Undifferentiated cells were removed with a pulse of 10 µg/ml mitomycin-C for 1 hr at day 14. Motor neurons were subsequently maintained in N2B27 media supplemented with 10 ng/ml BDNF and GDNF, with half media changes occurring twice per week.

TDP43-GFP mislocalization was induced via transduction with AAVs expressing anti-GFP nanobodies. In our TDP43-GFP-NES/ CTRL cell line, TDP43-GFP mislocalisation was achieved via addition of 1 µg/mL doxycycline, which was replenished every 48 hr.

## Immunocytochemistry

Cells were fixed in 4% paraformaldehyde for 20 min at room temperature (RT), followed by permeabilization in ice-cold methanol for 5 min. Blocking was performed in 1% BSA (in PBS) for 1 hr (RT), and primary antibodies (see Key resources table) were incubated overnight at 4°C. Next day, wells were washed in PBS. Secondary antibodies (Molecular Probes, 1:2000) were incubated for 1–2 hr (RT), and nuclei were stained with Hoechst 33542 (1:1000; Molecular Probes). All antibodies were diluted in the blocking agent. Plates were imaged using ImageXpress Pico (Molecular Devices). Super-resolution images were captured using the Zeiss LSM880 Confocal with Airyscan.

## Phenotypic analysis

Image analysis was conducted using custom scripts in CellProfiler (*Stirling et al., 2021*). To ensure accurate dendritic complexity measurements, we discarded soma that displayed overlapping dendrites from different neurons to prevent confounding effects. For CC3 analysis, we chose soma that did not overlap and were separated by at least 10 pixels, as CC3 intensities were quantified specifically within the soma. This adjustment allowed for a larger number of cells to be included in the CC3 analysis compared to the morphological analysis.

## RNA sequencing and analysis

RNA was extracted using the QIAzol and the NEB Monarch RNA extraction kit and ribosomal RNA was depleted using the NEBNext rRNA depletion kit (NEB). Libraries for Illumina short-read sequencing were prepared using the NEBNext Ultra II RNA library kit (NEB) according to the manufacturer instructions. Sequencing was performed at the Exeter sequencing centre using the NovaSeq platform. Reads were mapped to the human genome assembly hg19 using STAR (*Dobin et al., 2013*), and counts for each gene per sample were generated using the R package featureCounts (*Liao et al., 2014*). Differential expression analysis was performed using DESeq2 (*Love et al., 2014*). Alternatively spliced transcripts were detected using Leafcutter (*Li et al., 2018*). Libraries for long-read ONT platforms were prepared according to the manufacturer's instructions and sequenced on a PromethION sequencer at the Exeter sequencing centre. Reads were mapped to hg19 using minimap2 (*Li, 2018*), isoform counts were generated using FLAIR (*Tang et al., 2020*), and transcript isoforms were analyzed using the FICLE pipeline (*Leung et al., 2023*). Small RNA sequencing was performed commercially through Macrogen and the fastq data were mapped and counts generated using the miRDeep2 package (*Friedländer et al., 2008*). Differential expression was performed using DESeq2.

## RT-qPCR

Following lysis in QIAzol, total RNA was purified using the miRNeasy Micro Kit (Qiagen). A total of 500 ng RNA was used for the generation of cDNA using the High-capacity cDNA kit (Thermo Fisher). The RT-qPCR was performed using the Promega 2 x Master Mix and the QuantStudio 6 (Applied Biosystems) for 40 cycles. Housekeeping genes *GAPDH, HPRT1*, and *RPL13* were used for normalization. Using the ΔΔCt method, fold changes were calculated per sample relative to their appropriate control. A maximum Ct value of 33 was selected for all samples. Significance testing was performed using the Student's T-test. Primer sequences have been included in *Table 1*.

## Western blotting

Motor neuron samples at D40 were lysed in ice-cold RIPA buffer supplemented with HALT protease phosphatase inhibitors. Lysates were centrifuged at 500 × g, for 10 min at 4°C to remove cellular debris. The supernatant was used as the total protein fraction. A total of 3 µg protein was separated by SDS-PAGE gel electrophoresis (Bio-Rad) and transferred to 0.2 µm nitrocellulose membrane (Amersham Protran) for immunoblotting. Membranes were blocked for 1 hr at RT using the Intercept-T blocking buffer (Li-Cor). Primary antibodies (see Key resources table) were incubated overnight at 4°C. Secondary antibodies were incubated for 1 hr at RT. All antibodies were diluted in Intercept-T with

**Table 1.** Primer sequences used for RT-qPCR.

| Target | AS event | AS event loc. | Forward sequence | Reverse sequence |
| --- | --- | --- | --- | --- |
| UNC13A | canonical transcript | | GGACGTGTGGTACAACCTGG | GTGTACTGGACATGGTACGGG |
| UNC13A-CE | cryptic exon | intron 19 | TGGATGGAGAGATGGAACCT | GGGCTGTCTCATCGTAGTAAAC |
| STMN2 | canonical transcript | | AGCTGTCCATGCTGTCACTG | GGTGGCTTCAAGATCAGCTC |
| STMN2 trunc | Cryptic exon | intron 1 | GGACTCGGCAGAAGACCTTC | GCAGGCTGTCTGTCTCTCTC |
| ELAVL3 | canonical transcript | | TGCAGACAAAGCCATCAACACCC | GCTGACGTACAGGTTAGCATCC |
| ELAVL3-CE | cryptic exon | intron 3 | CCTGCTCTGAGGGATTGAGT | GTACAGGTTAGCATCCCGGA |
| PFKP | canonical transcript | | AGGCAGTCATCGCCTTGCTAGA | ATCGCCTTCTGCACATCCTGAG |
| PFKP-CE | cryptic exon | intron 3 | CTACCAGGGCATGGTGGA | GGAGAGTGTCTCCAGCATCC |
| ACTL6B | canonical transcript | | CTTCCACATCGACACCAATGCC | CAGGTTTGGCTCAGACTTGACG |
| ACTL6B-CE | cryptic exon | intron 4 | CCTGGATCACACCTACAGCA | ACCCAGGAGTTCGAAACTAGC |
| IGSF21 | canonical transcript | | GTCTGGAGGAAAACCAGCAC | TCTTGGTGTCATCCAGGTCA |
| IGSF21-CE | cryptic exon | intron 2 | GCCTGCAGGAGGTGTTTATG | CTCTCGCTTGCGGTAGTTCT |
| CYFIP2 | canonical transcript | | ATGCCCTGGATTCTAACGGACC | CTTGGTCAGAGCATAGTAGGCG |
| CYFIP2-CE | cryptic exon | intron 25 | CTCCAAGGAACCATTCTCCA | TGAGATTTCTTTCAGGGCTCA |
| FEZ1 | canonical transcript | | CCACTGGTGAGTCTGGATGA | TGGATCCCTCCAGTCTTCTG |
| FEZ1-CE | cryptic exon | intron 1 | TGGAGATGTGGGATGATGG | AGGGTCGAAGGTCCTCAAAC |
| CELF5 | canonical transcript | | CACCTACTGTGCCAGGGATT | ACTGTCCGCAGGCTTCAC |
| CELF5-IR | intron retention | intron 5 | CGTGAAGTTCTCCTCCCACA | CATCTGCACACACTCACACG |
| CACNA1E | canonical transcript | | CCATTGTCCATCACAACCAG | CCATGCCATACATCTTCAGG |
| CACNA1E-E19 | exon 15 inclusion | exon 19 | GAGGCCTTCAACCAGAAACA | CGACATGTGGTGTCTTCTCC |
| KCNQ2 | canonical transcript | | AGTACCCCCAGACCTGGAAC | TTCCGCCTCTTCTCAAAGTG |
| KCNQ2-E5 | exon 5 exclusion | exon 5 | ACGTCTTTGCCACATCTGC | CCCCTTCTCTGCCAAGTACA |
| TRAPPC12 | canonical transcript | | TCGTGGACAAGGAGAACCTCAC | GGACTTGTCTCTTTGCTGCAGC |
| TRAPPC12-CE | cryptic exon | intron 7 | ATGTGCAGCCCAAGTCAAG | TTGCCATGGAGTACATCACC |

0.2% Tween-20 and washed in TBST. Blots were imaged using the Li-Cor Odyssey Fc Imager. ImageJ (FIJI) was used for the quantification of band intensity. All bands were normalized to alpha-tubulin.

## Acknowledgements

We would like to thank Aaron Jeffries and the University of Exeter Sequencing Center for the Illumina and Oxford Nanopore Technologies long-read sequencing. Thanks to Jessica Board for her help with the TDP43 shRNAs. Thanks to Bradley Dyer and Corin Liddle from the Exeter Bioimaging Center for their help with the Zeiss LSM880 Airyscan imaging. This study was funded by an MNDA Pilot award and an MRC New Investigator Research Grant, and an MRC Response Mode Grant to AB. This study was supported by the National Institute for Health and Care Research Exeter Biomedical Research Centre. The views expressed are those of the author(s) and not necessarily those of the NIHR or the Department of Health and Social Care.

## Additional information

### Funding

| Funder | Grant reference number | Author |
|---|---|---|
| Medical Research Council | | Akshay Bhinge |
| Motor Neurone Disease Association | | Akshay Bhinge |
| NIHR BRC Exeter | | Jonathan Mill<br>Akshay Bhinge |

The funders had no role in study design, data collection and interpretation, or the decision to submit the work for publication.

### Author contributions

Johanna Ganssauge, Data curation, Formal analysis, Validation, Investigation, Visualization, Methodology, Writing – original draft, Writing – review and editing; Sophie Hawkins, Szi Kay Leung, Software, Formal analysis, Writing – review and editing; Seema Chandramohan Namboori, Validation, Methodology, Writing – review and editing; Jonathan Mill, Supervision, Writing – review and editing; Akshay Bhinge, Conceptualization, Data curation, Software, Formal analysis, Supervision, Funding acquisition, Validation, Investigation, Visualization, Methodology, Writing – original draft, Project administration, Writing – review and editing

### Author ORCIDs

Johanna Ganssauge ⓘ https://orcid.org/0000-0001-5150-5445
Sophie Hawkins ⓘ https://orcid.org/0000-0003-1977-5255
Szi Kay Leung ⓘ https://orcid.org/0000-0002-5607-4688
Jonathan Mill ⓘ https://orcid.org/0000-0003-1115-3224
Akshay Bhinge ⓘ https://orcid.org/0000-0003-2939-099X

Reviewer #2 (Public review): https://doi.org/10.7554/eLife.95062.3.sa1
Author response https://doi.org/10.7554/eLife.95062.3.sa2

## Additional files

### Supplementary files

Supplementary file 1. Excel file of per-replicate data points of RT-qPCR and WB data shown throughout the article.

MDAR checklist

### Data availability

The RNA-seq data is available on the Gene Expression Omnibus database with the following accession IDs: GSE290436, GSE290437, GSE290441. All qPCR and western blot analysis data have been included in the supplementary excel file. All data generated or analysed during this study are included in the manuscript and supporting files.

The following datasets were generated:

| Author(s) | Year | Dataset title | Dataset URL | Database and Identifier |
|---|---|---|---|---|
| Bhinge A, Ganssauge J, Namboori S, Leung S | 2025 | Rapid and Inducible Mislocalization of Endogenous TDP43 in a Novel Human Model of Amyotrophic Lateral Sclerosis | https://www.ncbi.nlm.nih.gov/geo/query/acc.cgi?acc=GSE290441 | NCBI Gene Expression Omnibus, GSE290441 |

*Continued on next page*

*Continued*

| Author(s) | Year | Dataset title | Dataset URL | Database and Identifier |
|---|---|---|---|---|
| Bhinge A, Ganssauge J, Namboori S, Leung S | 2025 | Rapid and Inducible Mislocalization of Endogenous TDP43 in a Novel Human Model of Amyotrophic Lateral Sclerosis | https://www.ncbi.nlm.nih.gov/geo/query/acc.cgi?acc=GSE290437 | NCBI Gene Expression Omnibus, GSE290437 |
| Bhinge A, Ganssauge J, Namboori S, Leung S | 2025 | Rapid and Inducible Mislocalization of Endogenous TDP43 in a Novel Human Model of Amyotrophic Lateral Sclerosis | https://www.ncbi.nlm.nih.gov/geo/query/acc.cgi?acc=GSE290436 | NCBI Gene Expression Omnibus, GSE290436 |

The following previously published datasets were used:

| Author(s) | Year | Dataset title | Dataset URL | Database and Identifier |
|---|---|---|---|---|
| Klim JR, Williams LA, Limone F, Davis-Dusenbery BN, Mordes DA, Burberry A, Steinbaugh MJ, Gamage KK, Kirchner R, Moccia R, Cassel S, Chen K, Wainger BJ, Wolff CJ, Eggan K, Juan GS | 2019 | ALS implicated protein TDP-43 sustains levels of STMN2 a mediator of motor neuron growth and repair | https://www.ncbi.nlm.nih.gov/geo/query/acc.cgi?acc=GSE121569 | NCBI Gene Expression Omnibus, GSE121569 |
| Fratta P | 2023 | ALS/FTD GWAS risk variant rs12973192 promotes severe cryptic splicing of the UNC13A transcript upon TDP-43 depletion | https://www.ebi.ac.uk/ena/browser/view/PRJEB42763 | EBI European Nucleotide Archive, PRJEB42763 |
| Liu EY, Russ J, Cali CP, Phan JM, Amlie-Wolf A, Lee EB | 2019 | Loss of Nuclear TDP-43 Is Associated with Decondensation of LINE Retrotransposons | https://www.ncbi.nlm.nih.gov/geo/query/acc.cgi?acc=GSE126542 | NCBI Gene Expression Omnibus, GSE126542 |

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
