## [Editor Report · eLife Assessment]

TDP-43 mislocalization is a key feature of some neurodegenerative diseases, but cellular models are lacking. The authors endogenously-tagged TDP-43 with a C-terminal GFP tag in human iPSCs, followed by expression of an intrabody-NES that targeted GFP to the cytosol. They **convincingly** report physical mislocalization and functional depletion of TDP-43, as measured by microscopy and RNAseq. This method will be **valuable** to investigators studying the biological consequences of TDP-43 mislocalization and the methodology is in line with the current state-of-the-art.

---

## [Referee Report · Reviewer #2 (Public review)]

Summary:

TDP-43 mislocalization occurs in nearly all of ALS, roughly half of FTD, and as a co-pathology in roughly half of AD cases. Both gain of function and loss of function mechanisms associated with this mislocalization likely contribute to disease pathogeneisis.

Here, the authors describe a new method to induce TDP-43 mislocalization in cellular models. They endogenously-tagged TDP-43 with a C-terminal GFP tag in human iPSCs. They then expressed an intrabody - fused with a nuclear export signal (NES) - that targeted GFP to the cytosol. Expression of this intrabody-NES in human iPSC derived neurons induced nuclear depletion of homozygous TDP-43-GFP, caused its mislocalization to the cytosol, and at least in some cells appeared to cause cytosolic aggregates. This mislocalization was accompanied by induction of cryptic exons in well characterized transcripts known to be regulated by TDP-43, a hallmark of functional TDP-43 loss and consistent with pathological nuclear TDP-43 depletion. Interestingly, in heterozygous TDP-43-GFP neurons, expression of intrabody-NES appeared to also induce the mislocalization of untagged TDP-43 in roughly half of the neurons, suggesting that this system can also be used to study effects on untagged endogenous TDP-43 as well as TDP-43-GFP fusion protein.

Strengths:

A clearer understanding of how TDP-43 mislocalization alters cellular function, as well as pathways that mitigate clearance of TDP-43 aggregates, is critical. But modeling TDP-43 mislocalization in disease-relevant cellular systems has proven to be challenging. High levels of overexpression of TDP-43 lacking an NES can drive endogenous TDP-43 mislocalization, but such overexpression has direct and artificial consequences on certain cellular features (e.g. altered exon skipping) not seen in diseased patients. Toxic small molecules such as MG132 and arsenite can induce TDP-43 mislocalization, but co-induce myriad additional cellular dysfunctions unrelated to TDP-43 or ALS. TDP-43 binding oligonucleotides can cause cytosolic mislocalization as well. Each system has pros and cons, and additional ways to induce TDP-43 mislocalization would be useful for the field. The method described in this manuscript could provide researchers with a powerful way to study the combined biology of cytosolic TDP-43 mislocalization and nuclear TDP-43 depletion, with additional temporal control that is lacking in current method. Indeed, the author see some evidence of differences in RNA splicing caused by pure TDP-43 depletion versus their induced mislocalization model. Finally, their method may be especially useful in determining how TDP-43 aggregates are cleared by cells, potentially revealing new biological pathways that could be therapeutically targeted.

Weaknesses:

The method and supporting data have some limitations.

• Tagging of TDP-43 with a bulky GFP tag may alter its normal physiological functions, for example, phase separation properties and functions within complex ribonucleoprotein complexes. The authors show that normal splicing function of GFP-TDP-43 is maintained, suggesting that physiology is largely preserved, but other functions and properties of TDP-43 that were not directly tested could be altered.

• Potential differences in splicing and micro RNAs between TDP-43 knockdown and TDP-43 mislocalization are potentially interesting. However, different patterns of dysregulated RNA splicing can occur at different levels of TDP-knockdown and can differ in different batches of experiments, thus it is difficult to asses whether the changes observed in this paper are due to mislocalization per se, or rather just reflect differences in nuclear TDP-43 abundance or batch effects.

---

## [Author Response]

The following is the authors’ response to the previous reviews

**Public Reviews:**

**Reviewer #1 (Public Review):**
Summary:Nuclear depletion and cytoplasmic mislocalization/aggregation of the DNA and RNA binding protein TDP-43 are pathological hallmarks of multiple neurodegenerative diseases. Prior work has demonstrated that depletion of TDP-43 from the nucleus leads to alterations in transcription and splicing. Conversely, cytoplasmic mislocalization/aggregation can contribute to toxicity by impairing mRNA transport and translation as well as miRNA dysregulation. However, to date, models of TDP-43 proteinopathy rely on artificial knockdown- or overexpression-based systems to evaluate either nuclear loss or cytoplasmic gain of function events independently. Few model systems authentically reproduce both nuclear depletion and cytoplasmic miscloalization/aggregation events. In this manuscript, the authors generate novel iPSC-based reagents to manipulate the localization of endogenous TDP-43. This is a valuable resource for the field to study pathological consequences of TDP-43 proteinopathy in a more endogenous and authentic setting. However, in the current manuscript, there are a number of weaknesses that should be addressed to further validate the ability of this model to replicate human disease pathology and demonstrate utility for future studies.Strengths:The primary strength of this paper is the development of a novel in vitro tool.Weaknesses:There are a number of weaknesses detailed below that should be addressed to thoroughly validate these new reagents as more authentic models of TDP-43 proteinopathy and demonstrate their utility for future investigations.(1) The authors should include images of their engineered TDP-43-GFP iPSC line to demonstrate TDP-43 localization without the addition of any nanobodies (perhaps immediately prior to addition of nanobodies). Additionally, it is unclear whether simply adding a GFP tag to endogenous TDP-43 impact its normal function (nuclear-cytoplasmic shuttling, regulation of transcription and splicing, mRNA transport etc).

We have included images of the untransduced day 20 MNs derived from the engineered TDP43-GFP iPSC lines and the unedited line (Supplementary Fig. 1B).

We acknowledge the reviewer’s concern about the potential impact of the GFP tag on TDP43's normal function. To address this, we have validated the functionality of TDP43 by assessing the inclusion of cryptic exons in highly sensitive targets such as *UNC13A* and *STMN2*, both of which are known to be directly regulated by TDP43.

We compared MNs derived from the unedited parent line with the TDP43-GFP MNs prior to nanobody addition. As measured by qPCR, cryptic exon inclusion in *UNC13A* and *STMN2* was not observed in the unedited or edited TDP43-GFP MNs (Supplementary Fig.1C), confirming that the tagging does not induce splicing defects by itself. The cryptic exon inclusion in *UNC13A* and *STMN2* were only observed in TDP43-GFP MNs expressing the NES nanobody (Supplementary Fig. 2D). These findings were further supported by our next-generation sequencing data, which also showed that cryptic exon inclusion was specific to the TDP43 mislocalization condition (Supplementary Fig.3 and 4).

Thus, we have strong evidence that the GFP-tagged TDP43 behaves similarly to the wild-type protein and does not interfere with its function in our model.

(2) Can the authors explain why there is a significant discrepancy in time points selected for nanobody transduction and immunostaining or cell lysis throughout Figure 1 and 2? This makes interpretation and overall assessment of the model challenging.

For the phenotypic data shown in Fig.1, we added the AAVs at day 18 or 20 and analyzed the cells at day 40. For the phosphorylated TDP43 western blot (revised Fig. 3D), cells were treated with doxycycline at day 20 to induce nanobody expression and samples were harvested at day 40. Thus, cells were harvested between days 20 or 22 after adding the nanobodies. The onset of transgene expression when using AAVs in neurons typically display slow kinetics. We observed TDP43 mislocalization in less than 50% of the neurons after 7 days post-transduction that peaked at 10-12 days after addition of the nanobodies, when more than 80% of the cells displayed TDP43 mislocalization. Hence, we do not believe that a two-day difference significantly alters the interpretation of the data.

The decision to harvest neurons at day 30 for the qPCR data was taken to investigate whether the splicing changes seen at day 40 from the transcriptomics analysis can be detected well before the phenotypes observed at day 40.

(3) The authors should further characterize their TDP-43 puncta. TDP-43 immunostaining is typically punctate so it is unclear if the puncta observed are physiologic or pathologic based on the analyses carried out in the current version of this manuscript. Additionally, do these puncta co-localize with stress granule markers or RNA transport granule markers? Are these puncta phosphorylated (which may be more reminiscent of end-stage pathologic observations in humans)?

We have tried immunostaining neurons for phosphorylated TDP43. However, our immunostaining attempts were unsuccessful. Depending on the antibody, we either saw no signal (antibody from Cosmo Bio, TIP-PTD-M01A) or even the control neurons displayed detectable phosphorylation within the nucleus (antibody from Proteintech 22309-1-AP). Consequently, we performed western blot analysis using an antibody from Cosmo Bio, (TIP-PTD-M01A) that clearly shows hyperphosphorylation of TDP43 in whole cell lysates (Fig. 3D, E). Hence, we have referred to these structures as puncta and not aggregates (Page 4).

To assess co-localization of the puncta with stress granules, we immunostained for the stress granule marker G3BP1. This was done in MNs that were treated with sodium arsenite (SA) or PBS as a control. In the PBS treated control MN cultures, TDP43 mislocalization alone did not induce stress granule formation. G3BP1+ stress granules were only observed following SA stress (0.5 mM, 60 minutes). Further, only a subset of TDP43 puncta overlapped with these stress granules (Supplementary Fig. 7) (Page 6).

(4) The authors should include multiple time points in their evaluation of TDP-43 loss of function events and aggregation. Does loss of function get worse over time? Is there a time course by which RNA misprocessing events emerge or does everything happen all at once? Does aggregation get worse over time? Do these neurons die at any point as a result of TDP-43 proteinopathy?

We agree that a time course to analyze TDP43 mislocalization and its consequences would be ideal. However, the mislocalization of TDP43 across neurons is not a coordinated process. At each given time instance, neurons display varying levels of TDP43 mislocalization. Answering the questions raised by the reviewer would require tracking individual neurons in real time in a controlled environment over weeks. Unfortunately, we currently do not have the hardware to run these experiments. However, we do observe increased levels of cleaved caspase 3 in MNs expressing the NES nanobody, indicating that these neurons indeed undergo apoptosis by day 40 (Fig.1).

We have, however, analyzed changes in splicing using qPCR for 12 genes over a time course starting as early as 4 hours after inducing mislocalization. We detect time-dependent cryptic splicing events in all genes as early as 8 hours after doxycycline addition, coinciding with the appearance TDP43 mislocalization (Fig. 4A, B).

(5) Can the authors please comment on whether or not their model is "tunable"? In real human disease, not every neuron displays complete nuclear depletion of TDP-43. Instead there is often a gradient of neurons with differing magnitudes of nuclear TDP-43 loss. Additionally, very few neurons (5-10%) harbor cytoplasmic TDP-43 aggregates at end-stage disease. These are all important considerations when developing a novel authentic and endogenous model of TDP-43 proteinopathy which the current manuscript fails to address.

As shown in Fig .1, the neurons expressing the NES-nanobody display a wide range of mislocalization as assessed by the % of nuclear TDP43 present. By titrating the amount of AAVs added to the culture, the model can be tuned to achieve a wide gradient of TDP43 mislocalization.

We calculated the size and percentage of neurons displaying TDP43 puncta. The size and the number of aggregates varies across the neurons that display TDP43 mislocalization. Around 50% of the neurons displayed small (1 um^2^) puncta while large puncta (> 5 um^2^) were observed in <10% of the cells, similar to observations in patient tissue (Fig. 1F).

**Reviewer #2 (Public Review):**
Summary:TDP-43 mislocalization occurs in nearly all of ALS, roughly half of FTD, and as a co-pathology in roughly half of AD cases. Both gain-of-function and loss-of-function mechanisms associated with this mislocalization likely contribute to disease pathogeneisis.Here, the authors describe a new method to induce TDP-43 mislocalization in cellular models. They endogenously tagged TDP-43 with a C-terminal GFP tag in human iPSCs. They then expressed an intrabody - fused with a nuclear export signal (NES) - that targeted GFP to the cytosol. Expression of this intrabody-NES in human iPSC-derived neurons induced nuclear depletion of homozygous TDP-43-GFP, caused its mislocalization to the cytosol, and at least in some cells appeared to cause cytosolic aggregates. This mislocalization was accompanied by induction of cryptic exons in well characterized transcripts known to be regulated by TDP-43, a hallmark of functional TDP-43 loss and consistent with pathological nuclear TDP-43 depletion. Interestingly, in heterozygous TDP-43-GFP neurons, expression of intrabody-NES appeared to also induce the mislocalization of untagged TDP-43 in roughly half of the neurons, suggesting that this system can also be used to study effects on untagged endogenous TDP-43 as well as TDP-43-GFP fusion protein.Strengths:A clearer understanding of how TDP-43 mislocalization alters cellular function, as well as pathways that mitigate clearance of TDP-43 aggregates, is critical. But modeling TDP-43 mislocalization in disease-relevant cellular systems has proven to be challenging. High levels of overexpression of TDP-43 lacking an NES can drive endogenous TDP-43 mislocalization, but such overexpression has direct and artificial consequences on certain cellular features (e.g. altered exon skipping) not seen in diseased patients. Toxic small molecules such as MG132 and arsenite can induce TDP-43 mislocalization, but co-induce myriad additional cellular dysfunctions unrelated to TDP-43 or ALS. TDP-43 binding oligonucleotides can cause cytosolic mislocalization as well. Each system has pros and cons, and additional ways to induce TDP-43 mislocalization would be useful for the field. The method described in this manuscript could provide researchers with a powerful way to study the combined biology of cytosolic TDP-43 mislocalization and nuclear TDP-43 depletion, with additional temporal control that is lacking in current method. Indeed, the authors see some evidence of differences in RNA splicing caused by pure TDP-43 depletion versus their induced mislocalization model. Finally, their method may be especially useful in determining how TDP-43 aggregates are cleared by cells, potentially revealing new biological pathways that could be therapeutically targeted.Weaknesses:The method and supporting data have limitations in its current form, outlined below, and in its current form the findings are rather preliminary.(1) Tagging of TDP-43 with a bulky GFP tag may alter its normal physiological functions, for example phase separation properties and functions within complex ribonucleoprotein complexes. In addition, alternative isoforms of TDP-43 (e.g. "short" TDP-43, would not be GFP tagged and therefore these species would not be directly manipulatable or visualizable with the tools currently employed in the manuscript).

With reference to our answer above, we have confirmed using qPCR and RNA-seq analysis that adding a GFP tag to the C-terminus of TDP43 does not result in an appreciable loss of functionality. We do not observe any cryptic exon inclusion in *STMN2* and *UNC13A*. Cryptic exon inclusion in these genes, especially *STMN2*, has been recognized as a very sensitive indicator of TDP43 loss of function (Supplementary Fig 1C, Supplementary 2D, Fig. 3, Fig.4)

We acknowledge that truncated alternatively spliced versions of TDP43 will lose the GFP-tag and cannot be manipulated with our system. Since our GFP tag is positioned on the C-terminus, our system cannot manipulate these truncated fragments as the tag is lost in these isoforms. But these isoforms, if present, should be detectable using the Proteintech antibody against total TDP43, which recognizes N-terminal TDP43 epitopes. However, western blot analysis, even 20 days after inducing TDP43 mislocalization, showed no truncated fragments. This suggests that TDP43 mislocalization alone is insufficient to generate significant levels of truncated isoforms. We have added this section to the Limitations paragraph (page 9).

(2) The data regarding potential mislocalization of endogenous TDP-43 in the heterozygous TDP-43-GFP lines is especially intriguing and important, yet very little characterization was done. Does untagged TDP-43 co-aggregate with the tagged TDP-43? Is localization of TDP-43 immunostaining the same as the GFP signal in these cells?

The purpose of the heterozygous experiments was to see whether mislocalized TDP43 could potentially trap the untagged TDP43. If this was not the case, we would have seen a maximum of 50% of the TDP43 signal mislocalized to the cytoplasm. The fact that a sizeable proportion of cells had significantly higher levels of TDP43 loss from the nucleus, indicates that mislocalized TDP43 can indeed trap the untagged protein fraction. We used GFP immunostaining to identify the tagged TDP43 while an antibody against the endogenous TDP43 protein was used to detect total TDP43 levels. In the cells that show near complete loss of nuclear TDP43, the total TDP43 signal coincides with the GFP (tagged TDP43) signal. We are unable to distinguish the untagged fraction selectively as we do not have an antibody that can detect this directly.

But we agree with the reviewer that these observations need further detailed follow-up that we are unable to provide currently. Hence, we have removed this figure from the manuscript.

(3) The experiments in which dox was used to induce the nanobody-NES, then dox withdrawn to study potential longer-lasting or self-perpetuating inductions of aggregation is potentially interesting. However, the nanobody was only measured at the RNA level. We know that protein half lives can be very long in neurons, and therefore residual nanobody could be present at these delayed time points. The key measurement to make would be at the protein level of the nanobody if any conclusions are be made from this experiment.

The reviewer has highlighted an important point. To address this issue, we tagged the nanobodies with a V5 tag that allowed us to directly measure nanobody levels within cells. After Dox withdrawal, we indeed observed significant expression of the nanobody within cells even after two weeks of Dox withdrawal. Extending the time point to three weeks allowed complete loss of the nanobody in most neurons. However, in contrast to our observations at two weeks, this was accompanied by a reversal of TDP43 mislocalization in these neurons at three weeks (Fig. 5).

Surprisingly, in less than 10% of the neurons, we observed >80% of the total TDP43 still mislocalized to the cytoplasm, despite nearly undetectable levels of the nanobody. Super-resolution microscopy further revealed persistent cytoplasmic TDP43 in these neurons that did not overlap with residual nanobody signal. This suggests that in these neurons, the nanobody was no longer required to maintain TDP43 mislocalization (Fig. 5, page 7)

(4) Potential differences in splicing and microRNAs between TDP-43 knockdown and TDP-43 mislocalization are potentially interesting. However, different patterns of dysregulated RNA splicing can occur at different levels of TDP-knockdown, thus it is difficult to assess whether the changes observed in this paper are due to mislocalization per se, or rather just reflect differences in nuclear TDP-43 abundance.

This a fair point. It is possible that microRNA dysregulation might require a greater loss of nuclear TDP43 and maybe more resilient to TDP43 loss as compared to splicing. We have acknowledged this in the discussion section (page 9).

**Recommendations for the authors:**

**Reviewer #1 (Recommendations For The Authors):**
(1) It would be helpful to include nuclear vs cytoplasmic ratios of TDP-43 instead of simply "% nuclear TDP-43"

We have used % nuclear TDP43 as these values have biologically meaningful upper and lower bounds, which makes it easier to compare across experiments. We found that using a ratio of nuclear vs cytoplasmic TDP43 intensities displayed higher variability and a wider range.

We have re-labelled the y-axis as “% Nuclear TD43 / soma TDP43” to make our quantification clearer. The conversion from % nuclear TDP43 to N/C is straightforward. If the % nuclear TDP43 is X, then the N/C ratio can be calculated as X / (100-X). For example, a % nuclear TDP43 of 80% would amount to an N/C ratio of 80/20 = 4.

(2) The axis descriptions in Figure 1D are very unclear. While this is described better in the figure legend, it would be beneficial to have a more descriptive y-axis title in the figure (which may mean increasing the number of graphs).

Axis descriptions and figures changed as recommended.

(3) In Figure 1, the time points at which iPSNs were transduced with nanobody and/or fixed for immunostaining is somewhat inconsistent across all panels. This hinders interpretation of the figure as a whole. The authors should use same transduction and immunostaining time points for consistency or demonstrate that the same phenotype is observed regardless of transduction and immunostaining day as long as the time in between (time of nano body expression) is consistent. Subsequently, in Figure 2, a different set of time points is used.

Please see our response in the public comments above

(4) In Figure 1, please show individual data points for each independent differentiation to demonstrate the level of reproducibility from batch to batch.

Data points have been shown per replicate (Supplementary Fig. 2)

We have refined our approach for phenotypic analysis to improve consistency across different clones. Previously, we set thresholds on % nuclear TDP43 to distinguish MNs with nuclear versus mislocalized TDP43. This was done by ranking all cells based on % nuclear TDP43 and applying quantile-based thresholds—designating the top 25% as control and the bottom 25% as mislocalized, ensuring equal number of cells per category. However, we observed significant variability in thresholds across clones. For instance, the E8 clone had thresholds of 96% and 29%, while the E5 clone had 93% and 40%.

To address this, we reanalysed the data using a standardized three-bin approach:

(1) Control: MNs expressing the control nanobody.

(2) Low-Moderate Mislocalization: MNs expressing the NES nanobody with > 40% nuclear TDP43.

(3) Severe Mislocalization: MNs expressing the NES nanobody with < 40% nuclear TDP43.

This approach ensured a more reliable comparison of TDP43 mislocalization effects across experiments. The conclusions remain the same.

(5) In Figure 2, please show individual data points.

Data points for all the qPCR analyses in the paper have been included as a supplementary text file.

(6) In Figure 3, please show individual data points.

Data points for the western blot data have been included as a supplementary data file.

All other comments are within the public review.

**Reviewer #2 (Recommendations For The Authors):**
(1) In general more robust quantification of many of the described phenotypes are necessary. In particular, no apparent quantification of cytosolic mislocalization was performed in Figure 1, or quantification of mislocalization of Figure 3F. It is unclear in the western blot in Fig 1G if TDP-43 signal were normalized to total protein, and of note it seems that expression of the intrabody-NES reduced total proteins in the western blots that were shown. No quantification or measurement of the insoluble material was done or shown.

We have quantified cytosolic mislocalization of TDP43 (Fig. 1C). The y-axis indicates the total TDP43 signal observed in the nucleus as a percentage of the total signal observed in the soma (including the nucleus). This value has the advantage of ranging between 100% (perfectly nuclear) to 0% (complete nuclear loss). The boxplots indicate that expression of the NES-nanobody results in a range of cytosolic mislocalization with a median value around 40% of the TDP43 remaining in the nucleus.

Western blot data in previous Fig. 1G was normalized to alpha-tubulin. We were unable to get a good signal for the insoluble fraction. From the alpha-tubulin alone, it cannot be concluded that NES-nanobody results in a decrease in total protein levels. In the revised western blot for phosphorylated TDP43 (Fig. 3D, E), we have quantified total and phosphorylated TDP43. Here, we observe a six-fold increase in the levels of phosphorylated TDP43 without a significant change in total TDP43 protein levels.

To avoid potential mis-interpretation of our results, we have now removed the previous Fig. 1G.

(2) Additional images of nearly all microscopy data at higher magnifications would be required to better evaluate TDP-43 localization. Ideally including images for each channel in addition to merged images, and especially for key figures such as Figure 1B, 3B, 3F.

Better images have been provided.

(3) No control images were shown for Figure 1F and 3F. It is unclear what the bright punctate spots of cytoplasmic TDP-43 GFP signal represent. Are these true aggregates? If so, additional characterization would be required before such conclusions can be made, beyond the relatively superficial western blot analysis that was done in Figure 1.

Control images have now been provided (Figure 1E). As we mentioned above, immunostaining analysis to characterize whether the aggregates are phosphorylated failed to provide a clear signal. However, we have now confirmed that the mislocalized TDP43 is indeed hyper-phosphorylated (Figure 3D, E). We have acknowledged this in the main text, and have referred to these as puncta reminiscent of aggregates (Page 4, Page 6).